# You May Need both Good-GAN and Bad-GAN for Anomaly Detection

## Abstract

Generative adversarial nets (GAN) have been successfully adapted for anomaly detection, where end-to-end anomaly scoring by so-called Bad-GAN has shown promising results. A Bad-GAN generates pseudo anomalies at the low-density area of inlier distribution, and thus the inlier/outlier distinction can be approximated. However, the generated pseudo anomalies from existing Bad-GAN approaches may (1) converge to certain patterns with limited diversity, and (2) differ from the real anomalies, making the anomaly detection hard to generalize. In this work, we propose a new model called Taichi-GAN [1] to address the aforementioned issues of a conventional Bad-GAN. First, a new orthogonal loss is proposed to regularize the cosine distance of decentralized generated samples in a Bad-GAN. Second, we utilize few anomaly samples (when available) with a conventional GAN, i.e., so-called Good-GAN, to draw the generated pseudo anomalies closer to the real anomalies. Our Taichi-GAN incorporates Good-GAN and Bad-GAN in an adversarial manner; which generates pseudo anomalies that contributing to a more robust discriminator for anomaly scoring, and thus anomaly detection. Substantial improvements can be observed from our proposed model on multiple simulated and real-life anatomy detection tasks.

## 1 Introduction

Anomaly detection (AD), a.k.a. outlier detection, refers to detecting uncommon samples (usually heterogeneous) out of inlier distribution (ChandolaVarun et al., 2009; Pang et al., 2021a). AD can be performed either in an unsupervised manner when only inliers are involved during the training process (Schlegl et al., 2019; Ngo et al., 2019; Ruff et al., 2018), or in a semi-supervised manner when few labeled anomalies are available (Ruff et al., 2020b; Liznerski et al., 2021).

Generative adversarial nets (GAN) (Goodfellow et al., 2014) are considered the most effective generative models nowadays, which have also been applied for AD (Schlegl et al., 2017; 2019; Zenati et al., 2018b; Siddiquee et al., 2019; Perera et al., 2019). Likewise, one-class classification and its variants are often used as the backbone of AD approaches (Schölkopf et al., 2001; Ruff et al., 2018; Liznerski et al., 2021; Wu et al., 2019). Recently, the adversarial training and one-class classification have been leveraged together to discriminate inlier/outlier in an end-to-end manner (Zheng et al., 2019; Ngo et al., 2019). Such efforts share a similar spirit with the work introduced in Dai et al. (2017) and here we call them Bad-GANs. Instead of generating samples to match the training data distribution as a conventional GAN (Goodfellow et al., 2014) that we refer as Good-GAN in this paper, a Bad-GAN pushes its generated samples towards the peripheral area of the training data distribution and considers them as pseudo anomalies. Rather than defining anomaly scores based on 1) the residual between reconstructed and original samples such as AnoGAN series (Schlegl et al., 2017; 2019), or 2) existing one-class classification-based measures such as Deep SVDD (Ruff et al., 2018) and FCDD (Liznerski et al., 2021), Bad-GANs directly leverage the learned discriminator to identify unseen anomalies from normal cases.

Although Bad-GANs have shown great potentials, they suffer from several disadvantages. *Dis1*: the generated samples can converge to limited patterns rather than being heterogeneous as real anomalies;

---

[1]Taichi is a concept from traditional Chinese culture, which refers to the co-existence and well-balance of two components with different properties. We borrow this concept to indicate the relationship between "Bad-GAN" and "Good-GAN" in our model.

*Dis2*: it is hard to ensure generated samples to resemble real anomalies, while abundance of irrelevant pseudo anomalies may negatively affect the discrimination of real anomalies; *Dis3*: no application has been found to integrate labeled anomalies in prior-art Bad-GANs.

In this work, we propose a new Taichi-GAN to address aforementioned disadvantages. ***First***, based on a state-of-the-art Bad-GAN (FenceGAN (Ngo et al., 2019)) framework, an orthogonal loss is proposed to increase the angular diversity, which addresses *Dis1*, and partially relieves *Dis2* by increasing the overlap between generated pseudo anomalies and real anomalies. ***Second***, we are the first to incorporate Good-GAN within a Bad-GAN framework, where the Good-GAN contributes to generating pseudo anomalies guided by few real anomalies, which addresses *Dis2* and *Dis3*. Our model is termed Taichi-GAN, as the Good-GAN and Bad-GAN are integrated in a competing yet balanced manner to generate pseudo anomalies towards better AD. We illustrate our ideas with 2D synthetic data and validate the proposed model with five datasets including clinical applications.

## 2  RELATED WORKS

Shallow AD models have their limitations in dealing with high-dimensional data and large-scale datasets (Ruff et al., 2020b). Here we focus on the most related deep AD approaches.

**Reconstruction-error-based (REB) Model**. REB approaches have been widely applied to image-based AD, where auto-encoder (Sato et al., 2018; Baur et al., 2018; Chen & Konukoglu, 2018) or GAN (Schlegl et al., 2017; 2019; Zenati et al., 2018b; Siddiquee et al., 2019; Akcay et al., 2018; Zenati et al., 2018a) is typically used for reconstruction. REB methods assume that anomalies would not be well reconstructed when the model is only trained with normal samples. Therefore, the reconstruction errors can be used to evaluate the abnormality. AnoGAN (Schlegl et al., 2017) is a representative REB approach which learns a manifold of anatomical variability and detects the anomalies with reconstruction errors mainly on pixel level. F-AnoGAN (Schlegl et al., 2019) follow the same framework as AnoGAN with improved processing speed. Han et al. (2021) ensemble multiple GANs for AD. However, REB approaches are mainly designed for data synthesis rather than AD, which may limit their comprehension for heterogeneous anomalies.

**One-class Classification-based Measure (OCCBM)**. OCCBM approaches (Ruff et al., 2018; Wu et al., 2019; Ruff et al., 2020b; Chalapathy et al., 2018) learn one-class embedding from normal samples to measure the derivation of an unseen case from the inlier distribution, where the measurement is then served for AD purposes. Deep OCCBM models extend the classical one-class models, e.g., one-class SVM (Schölkopf et al., 2001) and SVDD (Tax & Duin, 2004), by extracting features via deep nets (Ruff et al., 2018; Liznerski et al., 2021). OCCBM methods have been deployed to sophisticated applications, e.g., detecting anomalous event in practical scenes (Wu et al., 2019).

**Bad-GANs**. The Bad-GAN idea (Dai et al., 2017) is introduced for semi-supervised learning in multi-class classification. Similar approaches are found in the scope of the AD such as Sabokrou et al. (2018); Lim et al. (2018); Zheng et al. (2019); Ngo et al. (2019). Such techniques are termed as Bad-GANs in this paper. The key idea of Bad-GANs is to learn a classifier (i.e., the discriminator) based on inliers and generated pseudo outliers during training so that it can distinguish unseen outliers and inliers during inference. Bad-GANs belong to the one-class classification family and can be trained end-to-end. In addition, it does not require a manually defined one-class classification measure as OCCBM models (Pang et al., 2021a); the Bad-GAN discriminator handles the measure.

**AD Models Utilizing Real Anomalies**. Deep SAD (Ruff et al., 2020b) and ESAD (Huang et al., 2020) extend the Deep SVDD (Ruff et al., 2018) to take advantage of labeled anomalies. Siddiquee et al. (2019) propose a REB model for AD by considering normal and anomalies as two "styles" via image-to-image translation techniques. Few-shot learning and reinforcement learning are adapted to utilize limited label information, respectively (Tian et al., 2020; Pang et al., 2021b).

## 3  METHOD

### 3.1  BACKGROUND: GOOD-GAN AND BAD-GAN

GANs (Goodfellow et al., 2014) train generative models using a minimax game, which generate data instances from a noise distribution $z \sim p_z$ to match the given data distribution $p_{data}$ by iteratively

optimizing a generator $G$ and a discriminator $D$ with the following objective:

$$\min_G \max_D V(D, G) = \mathbb{E}_{x \sim p_{data}}[\log D(x)] + \mathbb{E}_{z \sim p_z}[\log(1 - D(G(z)))] \tag{1}$$

Dai et al. (2017) introduce the concept of the Bad-GAN that generates samples at the periphery of the training data. The generator is optimized by minimizing $\mathbb{E}_{x \sim p_G} \log p_{data}(x) \mathbb{I}[p_{data}(x) > \epsilon]$, where $\mathbb{I}[x]$ is an indicator function. With a threshold $\epsilon \in (0, 1)$, the training process penalizes the high-density samples and leaves the low-density ones unaffected, where the data distribution $p_{data}$ of Dai et al. (2017) is estimated with an additional model PixelCNN++ (Salimans et al., 2017).

The more recent FenceGAN (Ngo et al., 2019) shares the similar spirit of Dai et al. (2017), which generates samples at the boundary of a normal distribution. FenceGAN gets rid of the additional intensity estimation model as in Dai et al. (2017), and thus formulates an end-to-end training process. The generator loss [2] on the batch with size $N$ is defined as:

$$L_G = \frac{\beta \cdot N}{\sum_{i=1}^{N}(||G(z_i) - \mu||)} + \mathbb{BCE}(D(G(z)), \alpha) \tag{2}$$

where $\mathbb{BCE}(A, B)$ is the binary cross-entropy between $A$ and $B$. The first term is the dispersion loss which regularizes generated samples to be distant from their center $\mu$. The hyperparameter $\alpha$ is a discriminative anomaly score that pushes generated pseudo anomalies on inlier distribution boundary and $\beta$ is the so-called dispersion hyperparameter. The discriminator loss is defined as:

$$L_D = \mathbb{BCE}(D(x), \mathbf{0}) + \gamma \mathbb{BCE}(D(G(z)), \mathbf{1}) \tag{3}$$

where $\gamma \in (0, 1]$ is the so-called anomaly hyperparameter. The discriminator is used to compute an anomaly score ranged (0,1) for each unseen case during the inference stage.

## 3.2 ORTHOGONAL LOSS

As anything not normal is defined as an anomaly and anomalies are heterogeneous (Ruff et al., 2020b), we hypothesize that generated pseudo anomalies in a Bad-GAN framework should have high diversity to cover as many abnormal patterns as possible. Dispersion loss (Ngo et al., 2019) ($L_d$ in Fig. 1) regularizes generated pseudo anomalies lying at boundary of inlier distribution. Here, we propose to push pseudo anomalies further distributed to evenly cover the entire angular space.

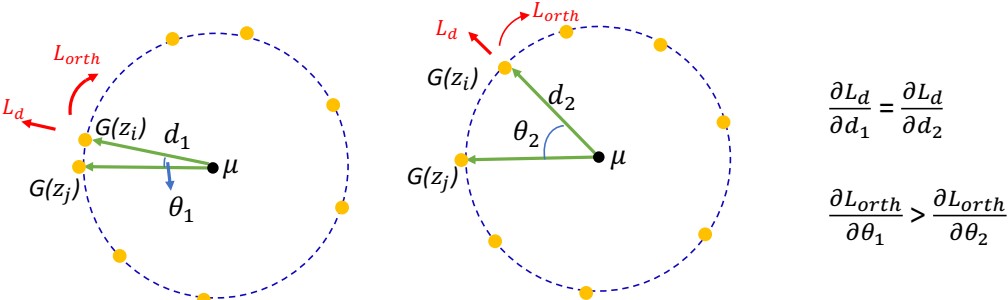

Figure 1: Motivation of our orthogonal loss. Two situations are illustrated, where $G(z_i)$ and $G(z_j)$ are a pair of generated pseudo anomalies. Indicated by right formulas, 1) the euclidean-distance-based loss $L_d$ e.g., dispersion loss (Ngo et al., 2019), stay the same for $G(z_i)$ given $d_1 = d_2$; and 2) the orthogonal loss $L_{orth}$ penalizes more on $G(z_i)$ given a smaller angle between $(G(z_i), G(z_j))$.

As verified in empirical experiments, the generated samples from a Bad-GAN can be prone to specific patterns, even with the Euclidean dispersion regularization of FenceGAN (Ngo et al., 2019). We introduce an orthogonal loss $L_{orth}$ to expand generated samples from the angular perspective. As depicted in Fig. 1, paired samples with smaller angles (left in Fig. 1) to each other can have a

---

[2]Equations are adapted from the official Github of FenceGAN, which have a different form with its paper.

larger penalty given the orthogonal regularization. The orthogonal regularization has been used for weight initialization and optimized on deep features (Brock et al., 2017; Arjovsky et al., 2015; Zhao et al., 2016). Here, we define our $L_{orth}$ in a decentralized way and optimized on generated pseudo anomalies in an AD framework:

$$L_{orth} = \frac{1}{N(N-1)} \sum_{i=1}^{N} \sum_{j \neq i} \left( \frac{(G(z_i) - \mu)^T (G(z_j) - \mu)}{||G(z_i) - \mu|| \cdot ||G(z_j) - \mu||} \right)^2 \qquad (4)$$

Instead of optimizing on the commonly used *Euclidean distance* and *individual regularization* (e.g., the dispersion term of Ngo et al. (2019) and the losses in OCCBM models (Ruff et al., 2018; 2020a;b)), our proposed orthogonal loss leverages *Cosine distance* and *relational regularization*[3] to be complementary to the dispersion term in the Bad-GAN framework.

### 3.3 THE INTUITION OF ADDING GOOD-GAN TO BAD-GAN

As Good-GANs optimize generated samples to match the training distribution, given few real anomalies are available, we take this Good-GAN nature to guide pseudo anomaly generation in AD tasks. Simultaneously, to avoid generated samples overfitting to limited anomaly patterns, we balance the Good-GAN and the Bad-GAN (including $L_{orth}$) with a hyperparameter $\delta$. Specifically, the Good-GAN guides the generated pseudo anomalies to resemble the reference distribution from given real anomalies, while the Bad-GAN regularizes pseudo anomalies at the boundary of inlier distribution. The Good-GAN and the Bad-GAN share the same generator, and have their own discriminators separately. The overall objective is defined as follows:

$$\min_{G} \max_{D_{bd}} \max_{D_{gd}} \{ \underbrace{V_{bd}(G, D_{bd})}_{\text{Bad-GAN}} + \underbrace{L_{orth}}_{\text{angular diversity}} + \underbrace{\delta V_{gd}(G, D_{gd})}_{\text{Good-GAN}} \} \qquad (5)$$

where $V_{bd}$ and $V_{gd}$ are the objective functions of the Bad-GAN and Good-GAN, respectively. $G$ is the shared generator. $D_{bd}$ and $D_{gd}$ are discriminators of Bad-GAN and Good-GAN, respectively. The generated samples serve as pseudo anomalies to train a classifier (i.e., $D_{bd}$) to discriminate unseen real anomalies from inliers during inference. $L_{orth}$ is the proposed orthogonal loss to further increase the diversities of pseudo anomalies. The overall model (Eq. 5) is termed as Taichi-GAN.

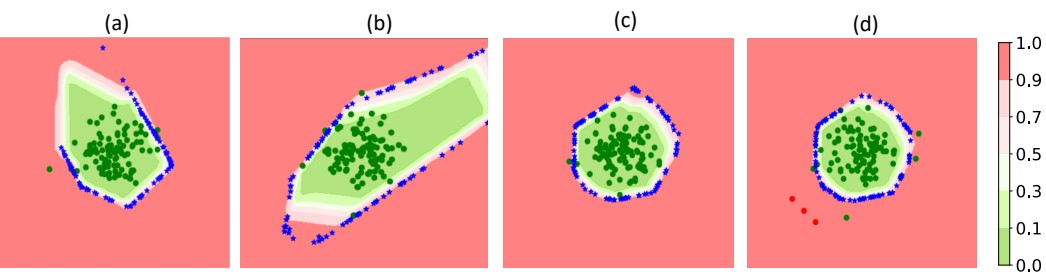

Figure 2: Experiments conducted on 2D synthetic data where $x_n \sim N((20, 20), 3)$, $z_a \sim N((0, 0), 8)$, $x_a = G(z_a)$. **(a)** Bad-GAN, **(b)** Bad-GAN + pull-away term ((Zhao et al., 2016), an orthogonal regularization without decentralizing samples), **(c)** Bad-GAN + $L_{orth}$, **(d)** Taichi-GAN (Bad-GAN + Good-GAN + $L_{orth}$). The green, red, and blue dots represent real inliers, reall outliers, and generated pseudo outliers, respectively. We use $\alpha = 0.9$ in Bad-GAN (here real anomalies score 1) following the setting in FenceGAN for MNIST. To illustrate the effect of a Good-GAN, we include three anomaly samples $\{(10, 10), (12, 8), (8, 12)\}$ to illustrate the efficacy of angular diversity and a Good-GAN in (d). The bar on the right encodes the anomaly score in color for reference.

As shown in Fig. 2(a), the Bad-GAN generates samples along the boundary of the inlier distribution, however, with limited diversity, even including a dispersion loss on Euclidean distance space. Adding

---

[3]Concepts of individual regularization and relational regularization have been introduced in Liu et al. (2021).

a pull-away term (Zhao et al., 2016) (an orthogonal regularization without decentralization to the sample average) is also not effective in this AD framework as in Fig. 2(b). The generated samples are more evenly distributed with the proposed orthogonal loss (Fig. 2(c)), which are more likely to cover samples that resemble real anomalies even though no real anomaly is included for training. Further, when few labeled anomalies are available, the Good-GAN inclusion pushes the generated samples to have higher density near the given anomalies, meanwhile, $L_{orth}$ maintains high diversity to cover other potential anomaly patterns (Fig. 2(d)).

## 3.4 THE PROPOSED TAICHI-GAN

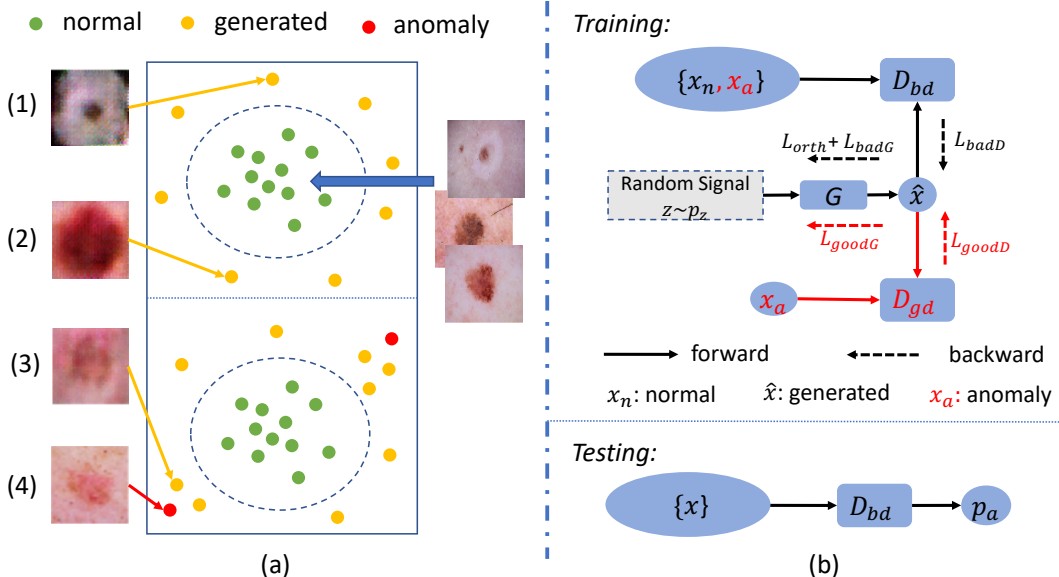

Figure 3: **(a)** Schematic visualization of the proposed method. The upper section illustrates the scenario without anomalies, which targets generating diverse samples. The bottom section includes few labeled anomalies (red dots), which targets generating samples under the guidance of few labeled anomalies while keeping high diversity. **(b)** Training (upper) and testing (bottom) framework. Under the hood of the entire Taichi-GAN framework, the Bad-GAN components are marked in black, while the additional Good-GAN components are in red. The discriminator of the Bad-GAN (i.e., $D_{bd}$) is used during the inference phase to score the anomalies.

Fig. 3(a) shows our motivation with inlier and (pseudo) outlier example from HAM10000 (Tschandl et al., 2018). The upper depicts the training with only normal samples, where we target generating highly diverse samples (e.g., (1) vs. (2) in Fig. 3(a)) at the boundary of normal distribution. The bottom shows the motivation on sample generation after adding few anomalies. Besides keeping the diversity, some generated samples are close to known anomalies (e.g., (3) is close to (4)).

We borrow Bad-GAN components from FenceGAN. The generator loss is defined as follows:

$$L_G = L_{orth} + \underbrace{\frac{\beta \cdot N}{\sum_{i=1}^{N}(||G(z_i) - \mu)||}} + \mathbb{BCE}(D_{bd}(G(z)), \alpha) + \underbrace{\delta \mathbb{BCE}(D_{gd}(G(z)), \mathbf{0})}_{L_{goodG}} \quad (6)$$
$$\underbrace{\phantom{\frac{\beta \cdot N}{\sum_{i=1}^{N}(||G(z_i) - \mu)||} + \mathbb{BCE}(D_{bd}(G(z)), \alpha)}}_{L_{badG}}$$

The generator losses of Bad-GAN ($L_{badG}$) and Good-GAN ($L_{goodG}$), is consistent with the form of Eq. 2 and shares the same spirit of Goodfellow et al. (2014), respectively. $G$ denotes the generator, $D_{bd}$ and $D_{gd}$ are the discriminators of Bad-GAN and Good-GAN, respectively. $L_{orth}$ is the proposed orthogonal loss to increase the diversity from the angular perspective.

The loss of Good-GAN discriminator is defined as: (Goodfellow et al., 2014):

$$L_{goodD} = \mathbb{BCE}(D_{gd}(x_a), \mathbf{0}) + \mathbb{BCE}(D_{gd}(G(z)), \mathbf{1}) \quad (7)$$

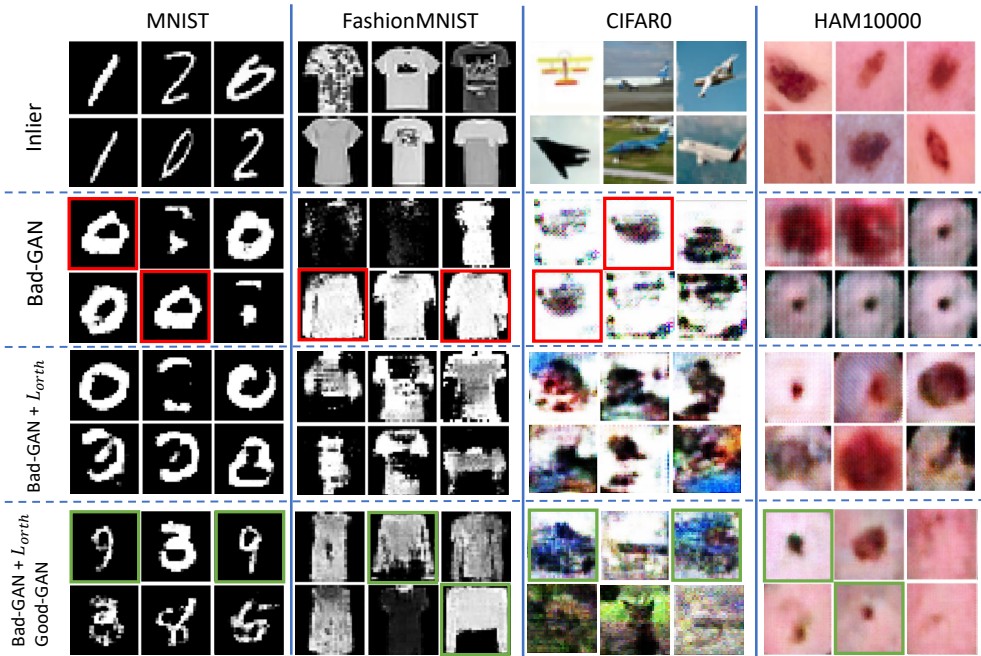

Figure 4: Generated pseudo anomalies among multiple datasets. The first row shows inlier examples. The second row shows samples generated only with Bad-GAN. The third row illustrates samples from Bad-GAN + $L_{orth}$ and the fourth row from the model of Bad-GAN + Good-GAN + $L_{orth}$, i.e., Taichi-GAN. Red boxes in the second row highlight examples clustering to specific patterns. Green boxes in the fourth row highlight cases that are close to the same anomaly class while keeping some extent of heterogeneity, even though only one sample per anomaly class is provided.

The loss of Bad-GAN discriminator is adapted from Ngo et al. (2019) by adding a third term:

$$L_{badD} = \mathbb{BCE}(D_{bd}(x_n), \mathbf{0}) + \gamma\mathbb{BCE}(D_{bd}(G(z)), \mathbf{1}) + \eta\mathbb{BCE}(D_{bd}(x_a), \mathbf{1}) \qquad (8)$$

where $x_n$ and $x_a$ are normal (inlier) and anomaly (outlier) cases, respectively. The third term includes real anomalies to train the discriminator of Bad-GAN. $\eta = 0$ when no anomalies are available. Note $V_{bd}$ and $V_{gd}$ (from Eq. 5) are composed of $\{L_{badG}, L_{badD}\}$ (from Eq. 6 and Eq. 8) and $\{L_{goodG}, L_{goodD}\}$ (from Eq. 6 and Eq. 7), respectively.

The procedures of Taichi-GAN are illustrated in Fig. 3(b). The unsupervised training setting (no real anomalies) is performed with only the black text components (Bad-GAN + $L_{orth}$). The red text components are added when few anomalies are available.

## 4 EXPERIMENTS

In addition to 2D synthetic data in Fig. 2, we involve five datasets to evaluate our method for AD. MNIST, Fashion-MNIST, and CIFAR10 are popular in previous literature (Ruff et al., 2020b; Ngo et al., 2019; Lee et al., 2018). HAM1000 and BraTS20 are medical datasets where the anomalies are well defined clinically. The Area Under the ROC Curve (AUC) (Fawcett, 2006) is used to evaluate the AD performance. The reported AUCs are average values over three independent runs of each experiment using different random seeds. Detailed network architectures, dataset splits, dataset sizes, model hyperparameters, and baseline implementations can be found in the Appendix A, B, and E.

### 4.1 DATA AND SETUP

**MNIST** (LeCun et al., 1998) contains 10 classes of digits (0-9). As one-vs-rest setting (i.e., one class is inlier and the rest classes are outlier) in MNIST is proved to be easy (Ruff et al., 2018; 2020b), we

consider the N-vs-rest setup. Class 0-2 are formalized as normal set and class 3-9 are formalized as anomaly set in our main setting (Table 1). Ablation studies considering different "N" and different amount of labeled anomalies are shown in Table 2 and Table 3, respectively.

**Fashion-MNIST** (Xiao et al., 2017) & **CIFAR10** (Krizhevsky, 2009) use one-vs-rest setting that we select class "T-shirt" and "Airplane" as the normal class and the rest as anomalies, respectively.

**HAM10000** (Tschandl et al., 2018) is a clinical-based dataset consisting of ~10000 dermatoscopic images with skin lesions of seven classes. The dominated class Melanocytic nevi ( >70%) is a common benign skin lesion which is set as inlier and the rest as outliers.

**BraTS20** (UPenn, 2020) is a recent version of BraTS challenge (Bakas, 2018). Three types of anomalies are included: the GD-enhancing tumor, the peritumoral edema, and the necrotic and non-enhancing tumor core. Here we simplify the problem to image-level disease detection motivated by (Siddiquee et al., 2019) in the AD setting. We extract the 2D slices from T2-FLAIR volumes, where slices with anomaly annotations are considered as outliers and the rest as inliers. Special handling and generated pseudo anomaly examples on BraTS20 can be found in Appendix D.

**Labeled Anomalies for Training**. We categorize our experiments on each of the five data collections into two folds, i.e., "No anomalies available" and "Few anomalies available" (see Table 1). For the latter fold, we include the following numbers of anomalies: *MNIST*, one sample per class in 3-9 classes; *Fashion-MNIST & CIFAR10*, one sample per class in classes except "T-shirt"; *HAM10000*, five samples per class except "Melanocytic nevi"; *BraTS20*, 300 slices with tumor from 5 scans.

**Methods for Comparison**. When no anomalies are available, the benchmark methods include OC-SVM (Schölkopf et al., 2001) (shallow AD model), Deep SSVD (Ruff et al., 2018), F-AnoGAN (Schlegl et al., 2019) (reconstruction-error-based model), original badGAN (Dai et al., 2017), and FCDD (Liznerski et al., 2021). When few anomalies are available, representative methods using outlier exposure or semi-supervised learning are included for comparison. FCDD+OE (Liznerski et al., 2021) integrates outlier exposure (OE) (Hendrycks et al., 2019) technique to utilize available real anomalies. Deep SAD (Ruff et al., 2020b) generalizes the Deep SVDD to semi-supervised setting. Two other naive approaches are included for benchmarking: (1) training a classification model using BCE regardless of limited anomalies, and (2) involving the available anomalies in the Bad-GAN discriminator training, i.e., Bad-GAN (semi).

**Variants of Taichi-GAN**. We perform experiments on the following settings: **1**) *Bad-GAN* only uses FenceGAN, **2**) *Bad-GAN+$L_{orth}$* includes orthogonal loss regularization, **3**) *Bad-GAN(semi)* trains the discriminator with few labeled anomalies besides the normal samples and generated samples, **4**) *Bad-GAN(semi)+Good-GAN* involves the Good-GAN training for sample generation on top of setting 3), **5**) *Taichi-GAN* is our ultimate approach, equivalent to *Bad-GAN(semi)+Good-GAN+$L_{orth}$*. Different settings serve as an approach-level ablation study.

## 4.2 BENCHMARK RESULTS

The AD performances on five datasets are reported in Table 1 across all benchmark methods and approach-level ablation settings described in Section 4.1. The proposed methods achieve the highest AUCs in both folds where *No* and *Few* anomalies available. Across all five datasets, we observe consistently that 1) the improvement from introducing orthogonal loss to Bad-GAN, 2) the improvement from leveraging few anomaly samples with a Good-GAN, and 3) Taichi-GAN including Good-GAN, Bad-GAN, and orthogonal loss achieves the best performances.

Generated pseudo anomalies are shown in Fig. 4 and appendix Fig. 6 (bottom left). With our orthogonal regularization, the generated samples are more diversified (3rd row vs. 2nd row). When few anomalies are available, by incorporating the Good-GAN, the generated samples look closer to real anomalies and maintain reasonable diversity (indicated by green boxes in the fourth row).

**Benchmark Analyses**. When no anomalies are available, increasing the diversity of generated samples with the proposed $L_{orth}$ effectively address *Dis1* (e.g., +5.6% AUC in MNIST and visual contrast of 3rd row vs. 2nd row in Fig. 4). Our proposed model achieves even better performance by integrating the Good-GAN to guide the pseudo anomaly generation in the Bad-GAN framework, and demonstrates the best effectiveness of consuming the limited labeled anomalies. By doing so, *Dis2* and *Dis3* are effectively addressed.

Table 1: AUCs (%) of benchmark comparison. "Bad-GAN" and "original BadGAN" refer to FenceGAN (Ngo et al., 2019) and the approach of Dai et al. (2017), respectively. Items with * are our settings and "Taichi-GAN" is our ultimate model. The highest AUCs are bold.

| Method | MNIST | FMNIST | CIFAR10 | HAM10000 | BraTS20 |
|---|---|---|---|---|---|
| No anomalies available | | | | | |
| OC-SVM (Schölkopf et al., 2001) | 75.98 | 86.53 | 61.60 | 57.11 | 85.01 |
| Deep SSVD (Ruff et al., 2018) | 86.13 | 89.28 | 68.31 | 78.42 | 83.08 |
| F-AnoGAN (Schlegl et al., 2019) | 86.49 | 87.65 | 71.10 | 80.60 | 83.52 |
| orignal BadGAN (Dai et al., 2017) | 83.70 | 88.13 | 69.81 | 80.29 | 88.84 |
| FCDD (Liznerski et al., 2021) | 77.60 | 73.72 | 68.10 | 74.85 | 79.30 |
| Bad-GAN (Ngo et al., 2019) | 86.52 | 87.85 | 68.01 | 79.77 | 88.64 |
| * Bad-GAN + $L_{orth}$ | **92.10** | **90.65** | **72.68** | **80.95** | **92.87** |
| Few anomalies are available (anomaly amount details in Section 4.1) | | | | | |
| FCDD+OE (Liznerski et al., 2021) | 92.13 | 91.58 | 76.23 | 79.42 | 90.92 |
| Deep SAD (Ruff et al., 2020b) | 96.95 | 93.11 | 73.61 | 81.51 | 84.13 |
| BCE | 92.82 | 93.50 | 78.21 | 81.39 | 90.46 |
| * Bad-GAN (semi) | 93.13 | 94.10 | 76.35 | 83.71 | 91.40 |
| * Bad-GAN (semi) + Good-GAN | 96.60 | 96.25 | 79.35 | 85.40 | 92.52 |
| * Taichi-GAN (**ultimate**) | **97.38** | **96.71** | **80.43** | **85.70** | **94.01** |

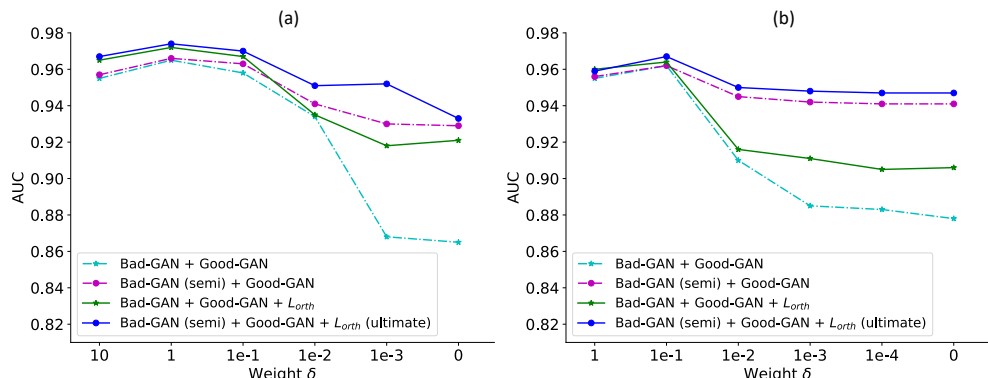

Figure 5: (a) Different weights of $\delta$ on MNIST. (b) Different weights of $\delta$ on Fashion-MNIST. $\delta = 0$ represents the Good-GAN is disfunctioned.

## 4.3 ABLATION STUDIES

In addition to the approach-level ablation study mentioned in **Variants of Taichi-GAN**, three more ablation studies have been conducted: 1) different weights $\delta$ between the Good-GAN and Bad-GAN, 2) different definitions of inlier/outlier, and 3) different availabilities of anomaly samples.

Fig. 5 shows results using different weights $\delta$ on MNIST and Fashion-MNIST. Note compared to *Bad-GAN (semi)*, the *Bad-GAN* represents that the real anomalies (even when available) are not included in training $D_{bd}$, thus the comparison between *Bad-GAN+Good-GAN* and *Bad-GAN* can better highlight the contribution of Good-GAN to AD tasks. Table 2 shows results of different definitions on inlier/outlier. The results of different available anomaly numbers/styles are shown in Table 3. Based on those ablation studies, we have further findings and analyses as follows:

**(1)** As in Fig. 5, both too large and too small $\delta$ decrease the performance of the Taichi-GAN. Generated pseudo anomalies can over-fit to few real anomalies (with limited patterns) with too large $\delta$. Contrastingly, too small $\delta$ decreases the efficacy of the Good-GAN (converge to Bad-GAN only).

**(2)** *Bad-GAN(semi)* includes real anomalies in training $D_{bd}$ where the discrimination performance can be considerably affected by real anomalies, which does not directly show the efficacy comparison of generated pseudo anomalies across methods. By contrast, the value of the Good-GAN are more

Table 2: AUCs (%) of different anomaly class (MNIST). For example, {0-4} represents 0-4 are outliers and the rest are inliers. 1 sample per outlier classes is available for bottom three settings.

| Anomaly Class | {5} | {0-2} | {0-4} | {0-6} |
|---|---|---|---|---|
| Bad-GAN | 90.21 | 86.49 | 73.41 | 66.72 |
| Bad-GAN + $L_{orth}$ | 91.23 | 92.12 | 75.67 | 75.42 |
| Bad-GAN(semi) | 94.72 | 93.13 | 84.02 | 75.94 |
| Bad-GAN(semi) + Good-GAN | 96.67 | 96.59 | 90.01 | 88.41 |
| Taichi-GAN (**ultimate**) | **96.90** | **97.41** | **90.12** | **88.73** |

Table 3: AUCs (%) of different anomaly availability where all 0-2 are inliers (MNIST). For example, [3-9]*5 represents 5 anomalies per 3-9 classes available.

| Available Anomalies | [3-6]*1 | [3-9]*1 | [3-9]*2 | [3-9]*5 | [3-9]*20 |
|---|---|---|---|---|---|
| BCE | 89.01 | 92.82 | 95.48 | 96.82 | 98.89 |
| Bad-GAN (semi) | 91.10 | 93.13 | 94.70 | 95.80 | 97.62 |
| Bad-GAN (semi) + Good-GAN | 93.70 | 96.60 | 96.80 | 97.42 | **99.11** |
| Taichi-GAN (**ultimate**) | **95.33** | **97.44** | **97.73** | **97.81** | 98.87 |

clearly highlighted in settings with Bad-GAN as in Fig. 5 (e.g., +9.0% AUC in MNIST when adding Good-GAN to Bad-GAN, i.e., $\delta = 0.1$ vs. $\delta = 0$ in *Bad-GAN+Good-GAN*).

(**3**) Results from Table 2 prove that our contributions of the orthogonal loss and the integration of Good-GAN make improvements to the baseline Bad-GAN consistently, while the degree of improvement varies in different settings and datasets (also supported by Table 1).

(**4**) When fewer anomalies are available (e.g., settings of "[3-6]*1" and "[3-9]*1" in Table 3), the performance gain from $L_{orth}$ and Good-GAN are more substantial (+ >4% AUCs).

(**5**) As in Table 3, with increasing number of real anomalies (e.g., "[3-9]*20"), we find 1) *Bad-GAN (semi)* can be less effective than *BCE*, and 2) the $L_{orth}$ does not necessarily improve the performance. This indicates when available anomalies can already cover a reasonable number of relevant patterns, the generated samples from pure Bad-GAN could introduce noise and thus decrease the discrimination power of $D_{bd}$. The $L_{orth}$ is built upon *Bad-GAN* by increasing the angular diversity of generated samples, and therefore also less effective in this scenario. This could be a limitation of the full-blown Taichi-GAN when sufficient diversity is already covered by the real anomalies. Yet, as a variant of Taichi-GAN, *Bad-GAN(semi)+Good-GAN*, remains most effective in this experiment.

## 5 CONCLUSION

In this paper, we address two key aspects in the Bad-GAN framework for AD that 1) generated pseudo anomalies from the Bad-GAN should have high diversity, 2) available anomalies (even few) can be used to guide pseudo anomaly generation. We propose a new Taichi-GAN that balances a "Good-GAN" with a "Bad-GAN" during training. Evaluated on five datasets, our model achieves the most promising performance in AD tasks compared with representative state-of-the-art methods. Instead of simply combine two types of GANs, Taichi-GAN value lies in the data efficiency as below.

**Further Discussion**. The key point of end-to-end anomaly scoring is to train an effective classifier (i.e., the discriminator of Bad-GAN) to distinguish unseen inliers and outliers, under the contexts that inliers are relatively sufficient while no or few real outliers are available during training. Therefore, the management of pseudo outlier generation is critical. When no outliers are available, the Bad-GAN is trained to generate peripheral samples of the inlier distribution as pseudo outliers. We validate in our experiments that higher sampling diversity achieved by the proposed orthogonal loss improves the quality of pseudo outliers towards a better classifier in inference. When few outliers are available, we demonstrate that the most effective way (compared with existing benchmarks with empirical experiments) of leveraging the available outliers is through our proposed Taichi-GAN, which enable the guidance from real outliers to the generation of pseudo outliers (e.g., the pseudo outlier boundary gets "thicker" near real outliers in Fig. 2(d)). We believe the core ideas of this work can be extended to other unsupervised or highly imbalanced classification tasks.

**Ethics Statement.** The concepts and information presented in this paper are based on research results that are not commercially available. Future commercial availability cannot be guaranteed.

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

## A    BASELINES AND HYPER-PARAMETERS

For baselines of benchmark Table 1, we adapt their official public code into our datasets/settings:

OC-SVM: https://github.com/lukasruff/Deep-SVDD

Deep SSVD: https://github.com/lukasruff/Deep-SVDD-PyTorch

Original BadGAN: https://github.com/kimiyoung/ssl_bad_gan

FCDD, FCDD + OE: https://github.com/liznerski/fcdd

Deep SAD: https://github.com/lukasruff/Deep-SAD-PyTorch

FenceGAN: https://github.com/phuccuongngo99/Fence_GAN

For the **Variants of Taichi-GAN** mentioned in Section 4.1, we share the same network structure and hyper-parameter for the fair comparison. We did not include additional data augmentation across all compared methods. Our experiments are conducted with PyTorch 1.1.

Our hyper-parameters are highly motivated from FenceGAN (Ngo et al., 2019), as we use FenceGAN as the "Bad-GAN" in our model. We follow the hyper-parameters of FenceGAN paper in MNIST, i.e., $\alpha = 0.9$, $\beta = 30$, $\gamma = 0.5$ for Eq.(6-9). Note that FenceGAN set $\alpha = 0.1$ for MNIST since the normal cases are defined as 0 and anomalies are defined as 1 in their paper, which is equivalent to $\alpha = 0.9$ in this paper. For the balance hyper-parameter $\delta$ between "Bad-GAN" and "Good-GAN", we set $\delta = 1$ for MNIST, $\delta = 0.1$ for Fashion-MNIST, HAM10000 and BraTS20. An ablation study of different $\delta$ has conducted in Section 4.3. If not specifically noted, the hyper-parameter $\eta$ is set to 0.1.

## B    DATA SPLITS

We use the same data splits and the same evaluation criteria for all compared methods for fair comparison. Note that the splits and criteria in some of original official Github are different, we have made necessarily adaptions about the data loader and evaluation checkpoint. The AUC results are reported on test sets. Note that all normal samples in the training set will be used for training, while no or few anomalies are included depending on settings. The general introduction of datasets can be found in Section 4.1. The following is about data splits:

**MNIST & Fashion-MNIST**: Original training set of MNIST contains 60000 samples from 10 classes. We randomly split the original training set as 90% / 10% for training and validation splits. We use the original test set as our test set, which contains 10000 samples. The AUCs reported on test set and the model selected based on the performance of validation set.

**CIFAR10**: Original training set of MNIST contains 50000 samples from 10 classes. We randomly split the original training set as 90% / 10% for training and validation splits. We use the original test set as our test set, which contains 10000 samples.

**HAM10000**: The HAM10000 consists of 10015 dermatoscopic images can be used for academic machine learning purposes. We split the HAM10000 dataset into training/validation/test split is 7001/1001/2013 samples for the AD task.

**BraTS20**: All the data in our experiments are from 369 patients of original BraTS20 training set. The training/validation/test splits are 12534/1351/2694 slices extracted from 280/30/59 patients.

## C USED METRIC DISCUSSION IN THIS PAPER

We use the Area Under receiver operating characteristic Curve (AUC) to evaluate the model's performance. We want to have an overall discrimination performance measurement to benchmark. AUC is known for the discrimination comparison across models and widely used in related references such as Deep SVDD and FCDD. When working on a specific application (which is not the target of this paper), for example, when there is a preference on less false positive rate or false negative rate, then precision / recall based metrics might be more preferred.

## D SPECIAL HANDLING OF BRATS20

Different from other three datasets, the intra-class variations can be larger than inter-class variations at pixel-level in BraTS20 (as the case in upper left Fig. 6), which increase the difficulty to learn a compact normal representation. Instead of feeding original image to AD detection models, we create processed different maps with F-AnoGAN (Schlegl et al., 2019). The advantages of creating the difference map include that all normal samples have the same ideal difference map (i.e., all-zero matrix), without the need to capture patterns of original images.

## E NETWORK STRUCTURES

We adapt the demonstration style of Liznerski et al. (2021) to show network structures in different datasets. Table 4, 5 show the generator and discriminator of MNIST and Fashion-MNST. Table 6, 7 show the generator and discriminator of HAM10000. Table 8, 9 show the generator and discriminator of BraTS20.

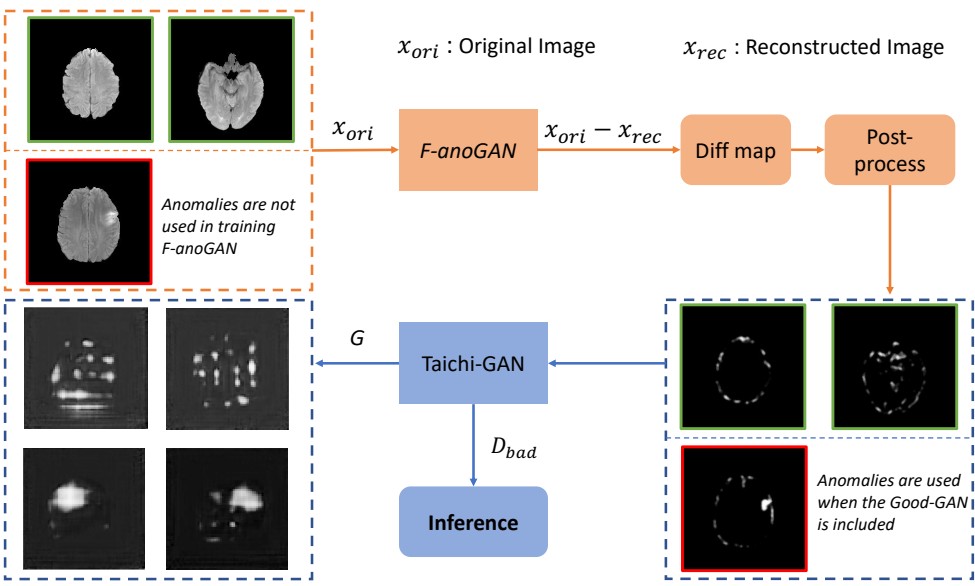

Figure 6: Anomaly detection pipeline for BraTS20. The upper left is examples of origin brain slices, i.e., two normal cases (upper) and one anomaly case (lower). Note that only normal cases are used for training F-anoGAN. Orange components (upper) are pre-processing steps for creating inputs (processed difference map) for the Taichi-GAN. The post-process includes three steps motivated from Baur et al. (2018): 1) threshoding with 0.3, 2) Gaussian Blur the image with kernel size $5 \times 5$ and standard deviation values $[1, 1]$, 3) median filter with kernel size $5 \times 5$. The right bottom is inputs of Taichi-GAN and the left bottom are generated pseudo anomalies by Taichi-GAN. In the inference stage, the anomaly score of each slice is obtained using $D_{bd}$.

Table 4: Network structures of generator for MNIST, Fashion-MNIST

| Layer (type) | Output Shape | Param # |
|---|---|---|
| Linear-1 | [-1, 256] | 51,456 |
| ReLU-2 | [-1, 256] | 0 |
| BatchNorm1d-3 | [-1, 256] | 512 |
| Linear-4 | [-1, 512] | 131,584 |
| ReLU-5 | [-1, 512] | 0 |
| BatchNorm1d-6 | [-1, 512] | 1024 |
| Linear-7 | [-1, 6274] | 3,217,538 |
| ReLU-8 | [-1, 6274] | 0 |
| BatchNorm1d-9 | [-1, 6274] | 12,544 |
| Reshape-10 | [-1, 128, 7, 7] | 0 |
| ConvTranspose2d-11 | [-1, 64, 14, 14] | 131,136 |
| ReLU-12 | [-1, 64, 14, 14] | 0 |
| BatchNorm2d-13 | [-1, 64, 14, 14] | 128 |
| ConvTranspose2d-14 | [-1, 1, 28, 28] | 1025 |
| Tanh-15 | [-1, 1, 28, 28] | 0 |
| Input shape: [-1, 200] | | |
| Total Params: 3.5M | | |
| Trainable Params: 3.5M | | |

Table 5: Network structures of discriminator for MNIST, Fashion-MNIST. The discriminator of "Bad-GAN" and "Good-GAN" share the same structure.

| Layer (type) | Output Shape | Param # |
|---|---|---|
| Conv2d-1 | [-1, 64, 14, 14] | 1088 |
| LeakyReLU-2 | [-1, 64, 14, 14] | 0 |
| Conv2d-3 | [-1, 64, 7, 7] | 65600 |
| LeakyReLU-4 | [-1, 64, 7, 7] | 0 |
| Reshape-5 | [-1, 3136] | 0 |
| Linear-6 | [-1, 512] | 1605632 |
| LeakyReLU-7 | [-1, 512] | 0 |
| Linear-8 | [-1, 128] | 655664 |
| LeakyReLU-7 | [-1, 128] | 0 |
| Linear-8 | [-1, 1] | 256 |
| Sigmoid-9 | [-1, 1] | 0 |

Input shape: [-1, 1, 28, 28]
Total Params: 1.7M
Trainable Params: 1.7M

Table 6: Network structures of generator for HAM10000

| Layer (type) | Output Shape | Param # |
|---|---|---|
| Linear-1 | [-1, 256] | 51,456 |
| ReLU-2 | [-1, 256] | 0 |
| BatchNorm1d-3 | [-1, 256] | 512 |
| Linear-4 | [-1, 512] | 131,584 |
| ReLU-5 | [-1, 512] | 0 |
| BatchNorm1d-6 | [-1, 512] | 1024 |
| Linear-7 | [-1, 4608] | 2363904 |
| ReLU-8 | [-1, 4608] | 0 |
| BatchNorm1d-9 | [-1, 4608] | 9216 |
| Reshape-10 | [-1, 128, 6, 6] | 0 |
| ConvTranspose2d-11 | [-1, 64, 12, 12] | 131,136 |
| ReLU-12 | [-1, 64, 12, 12] | 0 |
| BatchNorm2d-13 | [-1, 64, 12, 12] | 128 |
| ConvTranspose2d-14 | [-1, 64, 24, 24] | 65600 |
| ReLU-15 | [-1, 64, 24, 24] | 0 |
| BatchNorm2d-16 | [-1, 64, 24, 24] | 128 |
| ConvTranspose2d-17 | [-1, 3, 48, 48] | 3075 |
| Tanh-18 | [-1, 3, 48, 48] | 0 |

Input shape: [-1, 200]
Total Params: 2.8M
Trainable Params: 2.8M

Table 7: Network structures of discriminator for HAM10000. The discriminator of "Bad-GAN" and "Good-GAN" share the same structure.

| Layer (type) | Output Shape | Param # |
|---|---|---|
| Conv2d-1 | [-1, 64, 24, 24] | 3072 |
| LeakyReLU-2 | [-1, 64, 24, 24] | 0 |
| Conv2d-3 | [-1, 64, 12, 12] | 65600 |
| LeakyReLU-4 | [-1, 64, 12, 12] | 0 |
| Conv2d-5 | [-1, 64, 6, 6] | 65600 |
| LeakyReLU-6 | [-1, 64, 6, 6] | 0 |
| Reshape-7 | [-1, 2304] | 0 |
| Linear-8 | [-1, 512] | 1180160 |
| LeakyReLU-9 | [-1, 512] | 0 |
| Linear-10 | [-1, 128] | 655664 |
| LeakyReLU-11 | [-1, 128] | 0 |
| Linear-12 | [-1, 1] | 256 |
| Sigmoid-13 | [-1, 1] | 0 |

Input shape: [-1, 1, 48, 48]
Total Params: 1.4M
Trainable Params: 1.4M

Table 8: Network structures of generator for BraTS20

| Layer (type) | Output Shape | Param # |
|---|---|---|
| ConvTranspose2d-1 | [-1, 256, 6, 6] | 1638400 |
| BatchNorm2d-2 | [-1, 256, 6, 6] | 512 |
| ReLU-3 | [-1, 256, 6, 6] | 0 |
| ConvTranspose2d-4 | [-1, 128, 12, 12] | 819200 |
| BatchNorm2d-5 | [-1, 128, 12, 12] | 256 |
| ReLU-6 | [-1, 128, 12, 12] | 0 |
| ConvTranspose2d-7 | [-1, 64, 24, 24] | 204800 |
| BatchNorm2d-8 | [-1, 64, 24, 24] | 128 |
| ReLU-9 | [-1, 64, 24, 24] | 0 |
| ConvTranspose2d-10 | [-1, 32, 48, 48] | 51200 |
| BatchNorm2d-11 | [-1, 32, 48, 48] | 64 |
| ReLU-12 | [-1, 32, 48, 48] | 0 |
| ConvTranspose2d-13 | [-1, 32, 96, 96] | 25600 |
| BatchNorm2d-14 | [-1, 32, 96, 96] | 64 |
| ReLU-15 | [-1, 32, 96, 96] | 0 |
| Upsample-16 | [-1, 32, 192, 192] | 0 |
| Conv2d-17 | [-1, 1, 192, 192] | 800 |
| Tanh-18 | [-1, 1, 192, 192] | 0 |

Input shape: [-1, 256, 3,3]
Total Params: 2.7M
Trainable Params: 2.7M

Table 9: Network structures of discriminator for BraTS20. The discriminator of "Bad-GAN" and "Good-GAN" share the same structure.

| Layer (type) | Output Shape | Param # |
|---|---|---|
| Conv2d-1 | [-1, 32, 96, 96] | 832 |
| LeakyReLU-2 | [-1, 32, 96, 96] | 0 |
| Conv2d-3 | [-1, 64, 48, 48] | 51264 |
| InstanceNorm2d-4 | [-1, 64, 48, 48] | 128 |
| LeakyReLU-5 | [-1, 64, 48, 48] | 0 |
| Conv2d-6 | [-1, 64, 24, 24] | 102464 |
| InstanceNorm2d-7 | [-1, 64, 24, 24] | 128 |
| LeakyReLU-8 | [-1, 64, 24, 24] | 0 |
| Conv2d-9 | [-1, 128, 12, 12] | 204928 |
| InstanceNorm2d-10 | [-1, 128, 12, 12] | 256 |
| LeakyReLU-11 | [-1, 128, 12, 12] | 0 |
| Conv2d-12 | [-1, 256, 6, 6] | 819456 |
| InstanceNorm2d-13 | [-1, 128, 12, 12] | 512 |
| LeakyReLU-14 | [-1, 128, 12, 12] | 0 |
| Conv2d-15 | [-1, 256, 3, 3] | 1638656 |
| LeakyReLU-16 | [-1, 256, 3, 3] | 0 |
| Conv2d-17 | [-1, 1, 1, 1] | 2305 |
| Sigmoid-18 | [-1, 1, 1, 1] | 0 |

Input shape: [-1, 1, 192, 192]
Total Params: 2.8M
Trainable Params: 2.8M

