# OpenReview forum: "You May Need both Good-GAN and Bad-GAN for Anomaly Detection"
_ICLR.cc/2022/Conference — ICLR 2022 Submitted_

### Official Review · Reviewer_LRLn · 2021-10-31

**Correctness:** 3
**Technical Novelty And Significance:** 3
**Empirical Novelty And Significance:** 2
**Recommendation:** 5
**Confidence:** 4

**Main Review:**

# Strengths

The paper is organized well and easy to follow. The ideas to combine GANs of different quality is interesting and novel, however seems to be slightly limited as far as applying them to more complex benchmarks is concerned (see Weaknesses).

# Weaknesses

This manuscript has a couple of issues that need addressing. For one, the experimental evaluation is somewhat limited, in that it focuses on predominantly simple distributions and benchmarks (e.g. MNIST, FashionMNIST). I would recommend that the authors streamline their experiments for multi-modal problems, i.e. those that use multiple classes from MNIST (shown in Table 2 and 3).

In particular, the community has moved toward more demanding benchmarks, for example recent work proposes the use of semantic benchmarks for AD, see Ahmed & Courville (AAAI 2020), where 9 out of 10 classes are used as normal, and only 1 out of 10 is anomalous. Aligning experiments on MNIST with this challenging setting could potentially improve the paper's contribution, and make it a bit clearer how it compares to existing work. In addition, I would however strongly recommend that the proposed method is also evaluated on more complex benchmarks, e.g. CIFAR-10.

Next, the proposed method seems to require the availability of labeled anomalies. This is a very strong assumption for AD, and as "Rethinking Assumptions in Deep Anomaly Detection" of Ruff et al. (2021) clearly shows, often very simple classification-based methods that require only very few labeled anomalies achieve very robust performance. I am afraid this will make readers wonder why they should go through the trouble of training not one but two GANs to determine anomaly.

# Minor

There appears to be a formatting issue with the citation of Varun Chandola (2009).

Two GAN-based anomaly detection works which seem to be missing from this manuscript are:
- A. Berg, M. Felsberg, and J. Ahlberg. Unsupervised adversarial learning of anomaly detection in the wild. In European Conference on Artificial Intelligence, pages 1002–1008, 2020.
- L. Deecke, R. Vandermeulen, L. Ruff, S. Mandt, and M. Kloft. Image anomaly detection with generative adversarial networks. In Joint European Conference on Machine Learning and Knowledge Discovery in Databases, pages 3–17. Springer, 2018.

# Update

My rating has been increased slightly in response to a thorough revision, which now includes CIFAR-10 in the experiments of the manuscript. Concerns remain however with regards to performance versus competitors (such as F-AnoGAN), and because recent benchmarks like semantic AD have not been incorporated fully.

**Summary Of The Paper:**

The authors propose the coupling of coupling two GANs for anomaly detection: one which focuses on generating examples at the periphery of the distribution (hence the name BadGAN) and one that generates reference examples (the GoodGAN portion). These two generative models result from a joint optimization, and are evaluated on several anomaly detection benchmarks in this manuscript.


**Summary Of The Review:**

The paper presents an interesting idea that involves multiple GANs, but limits the evaluation to relatively simple benchmarks. Moreover, the proposed method relies on the presence of labeled anomalies, which are very difficult to obtain in practice.

---

> ### Author Response · Authors · 2021-11-21
> **Response to Reviewer LRLn**
>
> We thank the reviewer for the provided comments and hope the below can improve clarification.
>
> **C1. This manuscript has a couple of issues that need addressing. For one, the experimental evaluation is somewhat limited, in that it focuses on predominantly simple distributions and benchmarks (e.g. MNIST, FashionMNIST). I would recommend that the authors streamline their experiments for multi-modal problems, i.e. those that use multiple classes from MNIST (shown in Table 2 and 3).**
>
> Thank you. We agree that MNIST and FashionMNIST are relatively simple datasets, as they are straightforward to illustrate the proposed ideas. To challenge the proposed ideas, We also included two more complex clinical datasets (HAM10000 and BraTS20) for evaluation; the relatively low AUCs in Table 1 from benchmark methods indicate their complexity. In addition, as suggested by reviewers, we also added the experiments on CIFAR10 in the revision.
>
> As for “streamline experiments for multi-modal problems”, actually, Table 2 and 3 are ablation studies and we aim to show the sensitivity of our model to different experimental settings.
>
> **C2. the community has moved toward more demanding benchmarks, for example recent work proposes the use of semantic benchmarks for AD, see Ahmed & Courville (AAAI 2020), where 9 out of 10 classes are used as normal, and only 1 out of 10 is anomalous. Aligning experiments on MNIST with this challenging setting could potentially improve the paper's contribution, and make it a bit clearer how it compares to existing work. In addition, I would however strongly recommend that the proposed method is also evaluated on more complex benchmarks, e.g. CIFAR-10.**
>
> Thank you. In our ablation study of Table 2, we included the setting of one class as anomalies and the rest are normal in the first column of Table 2 (“5” as the outlier,  the remaining numbers as inliers), and the results indicate our model is generalizable at this setting. Moreover, since outliers are usually heterogeneous and inliers are relatively compact (ChandolaVarun et al., 2009; Pang et al., 2021a), it seems that the setting of  “one class as inlier and other classes as anomalies” makes more sense, which is indicated in Deep SSVD (Ruff et al., 2018), FCDD (Liznerski et al., 2021), Deep SAD (Ruff et al., 2020b), etc.
> Thanks for your recommendation, we have included the CIFAR-10 in this revision.
>
> **C3. The proposed method seems to require the availability of labeled anomalies. This is a very strong assumption for AD, and as "Rethinking Assumptions in Deep Anomaly Detection" of Ruff et al. (2021) clearly shows, often very simple classification-based methods that require only very few labeled anomalies achieve very robust performance. I am afraid this will make readers wonder why they should go through the trouble of training not one but two GANs to determine anomalies.**
>
> Actually, we categorize our experiments on each of the four data collections into two scenarios, i.e., “No anomalies available” and “Few anomalies available”. Our proposed models have superior results in both scenarios(Table 1-3).
> First, our experiments actually support the finding of Ruff et al. (2021), as shown in Table 1, only 1 sample per anomaly class is available for training (in total only 9 images), the AD AUC can reach 0.935 when only trained with binary cross-entropy loss (BCE). Our model can further improve the AUC to 0.967.
> Second, we are not only evaluating the “very simple” tasks. In the experiments of HAM10000 and BraTS20, the simple method such as BCE does not work that well, and our model provides consistently better efficiency of leveraging the few available anomalies. We also point out that our model is not simply to combine two types of GANs at item 5 of the overall summary.

---

> > ### Comment · Reviewer_LRLn · 2021-11-26
> > **Re: Response to Reviewer LRLn**
> >
> > Thanks to the authors for their response.
> >
> > Adding results for CIFAR-10 is a very useful addition. Nonetheless, I maintain that evaluating the semantic AD setting (Ahmed & Courville, AAAI 2020) would have been very interesting as well. One could expect GANs to struggle there (due to the complexity of the resulting normal distribution), so a positive result in this benchmark would have improved the quality of the paper further.
> >
> > All in all, I think the authors did a good job of addressing the general concerns that reviewers raised, and have increased my rating slightly in response to this. However there are several points which limit the potential impact of the submission, for example performances on CIFAR-10 appear very close to simpler methods like F-AnoGAN (Schlegl et al., 2019) and do not seem to match those reported in other recent works, e.g. Hendrycks et al. (NeurIPS 2019).

---

> > > ### Author Response · Authors · 2021-11-28
> > > **Second-round response to Reviewer LRLn**
> > >
> > > Thank the reviewer for increasing the rating and providing the second-round comments. I list our thoughts below.
> > >
> > > **Q: lack of semantic AD setting like (Ahmed & Courville, AAAI 2020)**
> > >
> > > **A**: Thanks for the reviewer’s recommendation and will cite this good paper (Ahmed & Courville, AAAI 2020) in a future revision. This paper studied the 1 anomaly vs. 9 normal settings. We have studied the 1 anomaly vs. 9 normal settings on MNIST in Table 2, along with other ablative settings. Further studies on more settings can be considered if necessary. Also, we have studied two clinical datasets targeting real clinical problems (not only simulation tasks on MNIST, FashionMNIST, CIFAR10, etc.), which should be more “semantic” based on our understanding. For example, for the BraTS20 dataset, the anomaly is considered as the image slices with the presence of brain tumors against the normal brain images.
> > >
> > > **Q: Performance is close to simpler methods like F-AnoGAN (Schlegl et al., 2019) and do not seem to match some recent works e.g. Hendrycks et al. (NeurIPS 2019)**
> > >
> > > **A**: We believe the paper Hendrycks et al. (NeurIPS 2019) mentioned by Reviewer LRLn is:
> > >
> > > *Hendrycks et al. Using Self-Supervised Learning Can Improve Model Robustness and Uncertainty (NeurIPS 2019)*
> > >
> > > We thank the reviewer’s paper recommendation and we will cite this good paper Hendrycks et al. (NeurIPS 2019) in a future revision.
> > >
> > > First, based on my understanding, F-AnoGAN is not a “simple” method, it is a representative model in reconstruction-error-based methods whose training process is not trivial. Each tested method may have highs and lows in performances across five benchmark datasets. F-AnoGAN may work relatively well on CIFAR10, where our proposed method in the unsupervised category shows small improvement. But across all benchmark datasets, the overall improvement from our method on top of F-AnoGAN is consistent and substantial.
> > >
> > > Second, we admit Hendrycks et al. (NeurIPS 2019) do have higher reported AUC values than ours in CIFAR10. Based on our understanding, this may result from the self-supervised learning or different backbones used in Hendrycks et al. (NeurIPS 2019). However, please note that in this manuscript, we mainly focus on the improvements from our proposed method on top of the baseline approaches (FenceGAN, Ngo et al. 2019) with the same backbone and hyper-parameters, as we showed with experiment results. We have not really pushed the limits of our proposed method yet. We can try integrating the training strategies from Hendrycks et al. (NeurIPS 2019) to further improve the performance, while such improvement will be less relevant to the major claims of the manuscript.

---

### Official Review · Reviewer_h6RB · 2021-11-01

**Correctness:** 3
**Technical Novelty And Significance:** 3
**Empirical Novelty And Significance:** 2
**Recommendation:** 3
**Confidence:** 4

**Main Review:**

Overall the paper introduces a new method that is of interest for the machine learning community.

Strengths:
- The paper is relevant to this conference and tackles an important research problem
- The related work is well presented in this paper and introduces well different anomaly detection methods
- The contributions of this work are motivated

However, my two main concerns are on the experimental methodology and on a less important manner on the writing.

Weaknesses/ Experiments:
- Dataset generations: it is unclear how the inlier/anomaly sets were selected; on MNIST Table 1, how would the methods perform for other combinations of "normal" sets and anomalies (than the 0-2 normal and 3-9 outliers)? on Fashion MNIST what does we "randomly" select the T-shirt as normal class mean? Would you attain similar results for other configurations? Why did you change these configurations for the ablation study ? On the labeled anomalies for training how did you decide to include 1 sample per class for MNIST/ Fashion MNIST and 5 per class for HAM10000, etc... ?
- Evaluation metrics: the authors provide results using AUCs (from an ROC curve, probably) of the anomaly detection methods, yet conventional methods in the literature also consider PRC curves using precision, recall; and eventually F1 scores. Depending on whether the focus is given on positive or negative classes (which may depend on datasets) in the anomaly detection task, both ROC and PRC curves could be discussed. This point is missing in the paper.
- Model selection: the method uses a set of hyper parameters to combine different losses (delta, Eq. 5; alpha, beta, gamma, etc. Eq. 2, 3; eta Eq. 8) and deep representations with network structures. Even though the paper relies on some previously picked models (information from Appendix A) the model selection procedure is *crucially* lacking in the paper. In Appendix B validation sets are mentioned but how are validated the models? This point is not discussed in the main paper but different hyperparameters are used for different datasets (Appendix A).
- Statistical significance: Were the experiments randomized and averaged (different random seeds) ?


Weaknesses/ Writing:
- The writing of the different disadvantages presented in the introduction (with the Dis notations) could be improved; especially Dis 3 seems an artefact of the Dis 2.
- The figures are not always clear; for e.g the orthogonal loss figure does not convey sufficiently the intuition of the orthogonal loss
- Evaluation metrics are not discussed enough; this is crucial in an empirical study paper.




**Summary Of The Paper:**

The authors of this paper propose a GAN-based method for the anomaly detection task. Their method relies on so-called Bad GANs (that uses the trained GAN discriminator to distinguish between inliers and pseudo-anomalies), a new orthogonal loss (that favorises generated samples to cover a larger angular space) and eventually a so-called Good GAN (which generator is shared with the Bad GAN; and discriminator is trained on "real" anomalies). The method is evaluated along a set of baselines in experiments that also include ablation studies.

**Summary Of The Review:**

The paper introduces a novel method that could be potentially interesting should the authors clarify all the concerns raised in the previous paragraph and provide extensive experiment results to address them. In the current state and given all the concerns on experimental methodology, I would recommend the rejection of the paper with regards to the standards of the ICLR conference.

UPDATE:

Please see the response I have provided to the authors. I decided to lower my score marginally and I encourage the authors to undertake the modifications suggested by the reviewers for future venues, especially on the empirical methodology and the writing. The paper in its current form may not be ready to my mind.

---

> ### Author Response · Authors · 2021-11-21
> **Responses to Reviewer h6RB**
>
> We thank the reviewer for the provided comments and we address them below.
>
> **Weakness 1. Dataset generations: it is unclear how the inlier/anomaly sets were selected; on MNIST Table 1, how would the methods perform for other combinations of "normal" sets and anomalies (than the 0-2 normal and 3-9 outliers)? on Fashion MNIST what does we "randomly" select the T-shirt as normal class mean? Would you attain similar results for other configurations? Why did you change these configurations for the ablation study? On the labeled anomalies for training how did you decide to include 1 sample per class for MNIST/ Fashion MNIST and 5 per class for HAM10000, etc... ?**
>
> (1) As we stated in Section 4, the one-vs-rest setting in MNIST is proved to be easy  (Ruff et al., 2018; 2020b) and we consider N-vs-rest set up in our main table (Table 1). Moreover, considering as an ablation study, we consider different numbers of “N” in the N-vs-rest setup (Table 1), including 1-vs-rest and 7-vs-rest.
>
> (2) Since Fashion-MNIST is harder than MNIST, we followed the popular 1-vs-rest setup as in   (Ruff et al., 2018; 2020b). “T-shirt” is the one class we picked randomly as inlier without picking other classes as prior.
>
> (3) For the ablation study, we want to evaluate if our model is robustly superior by changing the definition of inlier/outlier. That is why the configurations are changed.
>
> (4) Since the datasets of MNIST/Fashion-MNIST are less challenging than HAM10000, the inclusion of labeled anomalies is different in our setup. In addition, in the ablation study of Table 3. We also included experiments of different anomaly availability and have discussion in Section 4.3.
>
> **Weakness 2. Evaluation metrics: the authors provide results using AUCs (from an ROC curve, probably) of the anomaly detection methods, yet conventional methods in the literature also consider PRC curves using precision, recall; and eventually F1 scores. Depending on whether the focus is given on positive or negative classes (which may depend on datasets) in the anomaly detection task, both ROC and PRC curves could be discussed. This point is missing in the paper.**
>
> In this paper, we want to have an overall discrimination performance measurement to benchmark. AUC is known for the discrimination comparison across models and widely used in related references such as Deep SVDD and FCDD.  When working on a specific application (which is not the target of this paper), for example, when there is a preference on less false positive rate or false negative rate, then precision / recall based metrics might be more preferred.
>
> **Model selection: the method uses a set of hyper parameters to combine different losses (delta, Eq. 5; alpha, beta, gamma, etc. Eq. 2, 3; eta Eq. 8) and deep representations with network structures. Even though the paper relies on some previously picked models (information from Appendix A) the model selection procedure is crucially lacking in the paper. In Appendix B validation sets are mentioned but how are validated the models?**
>
> Thank you. We admit there are multiple hyper-parameters in our overall model. What we do is borrow the existing hyper-parameters in baseline models (Ngo et al., 2019) and study the sensitivity of our introduced hyper-parameters (i.e., $\eta$ and $\delta$).
> For $\delta$, we studied 6 different numbers and presented them in Figure 5. For $\eta$, we studied two numbers: $\eta$=0.1 for the model name with “(semi)” and $\eta$=0 for the model name without “(semi)”. We see the difference of performance between different $\eta$ is close to 0 at the optimal $\delta$. We also added the description of how the validation sets are used and basically different (our introduced) hyperparameters are dependent on the performance of the validation set.
>
> **Statistical significance: Were the experiments randomized and averaged (different random seeds)?**
>
> Thank you. Section 4: “The reported AUCs are average values over three independent runs of each experiment using different random seeds.”
>
> **Weaknesses/ Writing**
> **The writing of the different disadvantages presented in the introduction (with the Dis notations) could be improved; especially Dis 3 seems an artefact of the Dis 2.**
>
> Thank you. We agree that Dis2 and Dis3 are closely related but highlight different aspects of existing models.
>
> **The figures are not always clear; for e.g the orthogonal loss figure does not convey sufficiently the intuition of the orthogonal loss.**
>
> Basically, in Figure 1, we show that some existing methods regularize the diversity at Euclidean space and we propose a new loss focus on regularization on angular space.
>
> **Evaluation metrics are not discussed enough; this is crucial in an empirical study paper.**
>
> We have added the selection of evaluation metrics in the appendix.

---

> > ### Comment · Reviewer_h6RB · 2021-11-27
> > **Response**
> >
> > I would like to thank the authors for their response and their update.
> >
> > Unfortunately the authors failed to respond to the main concerns I had on the empirical methodology especially, on the combination of normal sets vs anormal (the N vs rest setup could be more detailed with further combinations than the ones in the main paper, at least in the Appendix for space reasons ) and the class that the authors themselves "picked randomly". I did not find the description/response to how the validation sets are used so as to address my concerns on the model selection procedure, especially on how to perform hyperparameter selection. This point is crucial for the AD task which is of interest for practitioners, if the authors find previous baselines to not discuss this point enough it might be helpful for the overall community to raise this issue, address it and not follow previous works.
> >
> > Overall and with regards to the other reviewers concerns (baselines/benchmark results asked and raised by Reviewer LRLn and 83KL), I think this paper can be improved from the empirical methodology (avoid conceptual overfitting), the baselines and the writing to make it clearer. I have decided to lower my score but encourage the authors to pursue their method and their work. This work has the potential to be a significant contribution for the AD task but in its current form it is not ready for publication at the ICLR.

---

> > > ### Author Response · Authors · 2021-11-28
> > > **Second-round response to Reviewer h6RB**
> > >
> > > Thank the reviewer for providing the second-round comments. I list our thoughts below.
> > >
> > > **Q: Concerns on empirical methodology**
> > >
> > > **A**: We try to demonstrate the generalization of the proposed method from two dimensions, the 1st dimension from different datasets as in Table 1, and the 2nd dimension from different combinations of normal sets vs. anomalies as Table 2. We consider Table 2 addressing the concerns raised by the reviewer, where we mainly studied MNIST through ablation studies including 1 abnormal vs. 9 normal, 3 abnormal vs. 7 normal, 5 abnormal vs. 5 normal, 7 abnormal vs. 3 normal. For now, we consider these two dimensions approximately cover the space of proving the robustness of the proposed method, while further experiments can be considered in future revision if necessary, for example, ablative studies on datasets other than MNIST (such as choice of anomaly class raised by the reviewer).
> > >
> > > **Q: Validation usage and hyper-parameter selection**
> > >
> > > **A**: Based on the 1st round of review, we included the sentence in the first-round response: “basically different (our introduced) hyperparameters are dependent on the performance of the validation set”  and have added the sentence in appendix B: “The AUCs reported on the test set and the model selected based on the best performance of validation set”. The criterion is applied across all datasets. The model selection with validation set includes the hyper-parameter selection (Figure 5) and epoch number. We hope these address the reviewer's concern.

---

### Official Review · Reviewer_83KL · 2021-11-02

**Correctness:** 4
**Technical Novelty And Significance:** 3
**Empirical Novelty And Significance:** 4
**Recommendation:** 6
**Confidence:** 3

**Main Review:**

**Pros**

1. The paper addresses how to generate pseudo anomalies to improve the anomaly detection. For me, the application of orthogonal loss into the problem itself looks simple but effective.

2. The proposed TaiChi-GAN, which combines Good-GAN, Bad-GAN and Orthogonal loss,  is novel for capturing inliers and generating better pseudo anomalies. Good-GAN is sort of the regularization to Bad-GAN regarding the learning of the anomaly distribution. The design for the framework is reasonable and interesting.

3. This paper provides comprehensive studies and experiments, including both qualitative analysis and quantitative results, to show the effectiveness of the proposed framework.

**Cons**

1. However, I have concerns regarding how Good-GAN learns to generate “pseudo anomalies to resemble the reference distribution from given real anomalies”. Not sure how the authors verify this claim, e.g., when looking at Fig. 2d?

2. It is good to have studies over new hyper-parameters introduced in the framework, e.g., \gamma, \eta.

3. MNIST and Fashion MNIST are not too challenging datasets, so it would be good to have experiments in e.g., CIFAR, SVHN like Bad-GAN and GAB-based anomaly papers?

4. May consider citing missing GAN works on anomaly detection (ideally comparing them) in the paper:
[1] DOPING: Generative Data Augmentation for Unsupervised Anomaly Detection with GAN
[2] Efficient GAN-Based Anomaly Detection
[3] GAN Ensemble for Anomaly Detection

**Summary Of The Paper:**

The paper proposes a new interesting method for anomaly detection. In particular, to overcome the limitations of existing Bad-GAN, the authors introduce the orthogonal loss to regularize the generation of anomaly samples to be distributed evenly at the periphery of the training data. Furthermore, in the scenario of available anomalies, the authors combine Bad-GAN and Good-GAN together, in which Good-GAN learns to generate the anomalies while Bad-GAN reguralizes the anomaly pseudo anomalies at the boundary of inlier distribution.

**Summary Of The Review:**

Overall, I vote for accepting. I like the idea of combining Good-GAN and Bad-GAN together, which is novel and interesting to me. The proposed orthogonal loss is simple and effective as being used as the complementary of the existing dispersion loss.  The results on benchmark datasets shown in the paper are encouraging and outperforms many existing works in both scenarios without or few anomalies observed.

---

> ### Author Response · Authors · 2021-11-21
> **Responses to Reviewer 83KL**
>
> We thank reviewers for the positive feedback and pointing out potential weaknesses. We address the comments below:
>
> **Cons.1. However, I have concerns regarding how Good-GAN learns to generate “pseudo anomalies to resemble the reference distribution from given real anomalies”. Not sure how the authors verify this claim, e.g., when looking at Fig. 2d?**
>
> In our design, the bad-GAN is to make the pseudo anomalies located at the boundary of inlier distribution and good-GAN is to push the pseudo anomalies to match (few) real anomalies. Those are two forces that guide the generation of anomalies. As shown in Fig. 2d, the generated blue dots still lie at the inlier distribution boundary but have a higher density near the real anomalies. As also supported by Figure 4 (last row),  generated samples are closer to the real anomalies while keeping some extent of heterogeneity, even though only one sample per anomaly class is provided.
>
> **Cons. 2. It is good to have studies over new hyper-parameters introduced in the framework, e.g., $\gamma$, $\eta$.**
> Thank you. Actually, the $\gamma$ is from the baseline bad-GAN model (Ngo et al., 2019) and we set it = 0.5 follows (Ngo et al., 2019). For \eta, we studied two cases, $\eta$=0.1 and $\eta$=0. In Figure 5, $\eta$=0.1 is applied when “(semi)” is included in the model name, including “Bad-GAN (semi) + Good-GAN” and “Bad-GAN (semi) + Good-GAN + $L_{orth}$”, otherwise, $\eta$ is set to 0. We see the difference of  $\eta$=0.1 vs. $\eta$=0 is nearly 0 at the optimal $\delta$, indicated by Figure 5(a) when $\delta$=1 and Figure 5(b) when $\delta$=0.1.
> We have clarified our statement in revision to prevent further confusion.
>
> **Cons. 3. MNIST and Fashion MNIST are not too challenging datasets, so it would be good to have experiments in e.g., CIFAR, SVHN like Bad-GAN and GAN-based anomaly papers?**
>
> Thank you, we have included the experiment of CIFAR10 in the revision.
> In addition, we would like to point out that we have two clinical datasets that are more complex than MNIST and Fashion MNIST, even CIFAR and SVHN.
>
> **Cons. 4. May consider citing missing GAN works on anomaly detection (ideally comparing them) in the paper: [1] DOPING: Generative Data Augmentation for Unsupervised Anomaly Detection with GAN [2] Efficient GAN-Based Anomaly Detection [3] GAN Ensemble for Anomaly Detection.**
>
> Thank you, we have cited the suggested works in the revision. [1] and [2] can be categorized into large groups of "bad GANs" and "reconstruction-based methods", respectively. We have included representative methods for comparison in those two categories. [3] is an ensemble-based model which is out of experiment comparison in this paper.

---

> > ### Comment · Reviewer_HJ5T · 2021-11-29
> > **Rebuttal Response**
> >
> > Including the CIFAR 10 results is definitely an improvement. Unfortunately the results are rather lackluster and there are unsupervised methods that beat all the methods included in the paper (including the semi-supervised versions). In particular "Using Self-Supervised Learning Can Improve Model Robustness and Uncertainty" by Hendrycks et al gets above 90AUC completely unsupervised. Regarding few samples, I'd recommend the authors look at "Rethinking Assumptions in Deep Anomaly Detection" by Ruff et al. Baseline performance on this problem is quite important since its a common baseline that seems to require deep methods to do well (one an achieve good MNIST performance with just a density estimator for example).
> >
> > In light of these points I don't think the paper is bad, but its results/theory/novelty aren't significant enough to recommend acceptance.

---

### Official Review · Reviewer_HJ5T · 2021-11-05

**Correctness:** 4
**Technical Novelty And Significance:** 2
**Empirical Novelty And Significance:** 2
**Recommendation:** 5
**Confidence:** 3

**Main Review:**

 * The problem addressed by the authors is important and GAN based AD does seem appropriate for certain datasets. Experimentally, a lot of the experimental improvements are quite small.
 * It would be interesting to see how the method performs on CIFAR10 one v rest, since that seems to be a bit of a standard dataset and fairly challenging for generative model based AD.
 * The paper lacks any theory.
 * The lack of theory is compounded by the rather small experimental setup. Lacking any theory a paper should have very strong experimental support.
 * Clarity is pretty good.
 * Good lit review.

Overall the paper is okay, I don't think the results are impressive enough, or that the proposed method is novel enough for a top tier conference.

**Summary Of The Paper:**

In this paper the authors propose a method for image anomaly detection (AD) based on GANs. The use the GAN discriminator based on a method called "Bad GAN" where one trains a GAN to produce low likelihood samples (trained using the nominal data) and then use the discriminator, which has been generalized to a large class of samples, to determine if a test sample is anomalous. In the proposed method the authors include $L_{orth}$ to enforce angular diversity of generated samples, a "good" gan discriminator that is trained to correctly determine if a sample is generated, and a bad gan discriminator inducing the generator to make points near the boundary of the nominal set distribution. At test time the "bad" discriminator used to determine an anomaly score. The authors test their method experimentally on F/MNIST and some medical datasets.

**Summary Of The Review:**

Overall the paper is okay, I don't think the results are impressive enough, or that the proposed method is novel enough for a top tier conference.

---

> ### Author Response · Authors · 2021-11-21
> **Responses to Reviewer HJ5T**
>
> We thank the reviewer for the provided comments and we address them below.
>
> **R1.C1. The problem addressed by the authors is important and GAN-based AD does seem appropriate for certain datasets. Experimentally, a lot of the experimental improvements are quite small.**
>
> Thanks for acknowledging our contributions. As for the experiments, we do show that our model brings large improvement over the compared baseline across five datasets. As shown in Table 1, take the MNIST for example, the introduction of $L_{orth}$ can increase nearly 6% AUC (Bad-GAN vs. Bad-GAN+$L_{orth}$).  The introduction of Good-GAN over a Bad-GAN framework with few anomalies increased 3.5% AUC values.
> Indeed, some comparisons as in Table 1, e.g., “ Bad-GAN(semi) + Good-GAN” vs. “ Taichi-GAN (ultimate)”, the improvements are quite small (indicated by reviewer). However, these are all variants of our methods, and small differences are expected. We also include the discussions in the ablation study section.
>
> **R1.C2. It would be interesting to see how the method performs on CIFAR10 one v rest, since that seems to be a bit of a standard dataset and fairly challenging for generative model-based AD.**
>
> We have included the experiments.
>
> **R1.C3. The paper lacks any theory.**
>
> Our clarification for novelty is stated in item 5 of the overall rebuttal summary.
>
> **R1.C4. The lack of theory is compounded by the rather small experimental setup. Lacking any theory a paper should have very strong experimental support.**
>
> We have illustrated our ideas with 2D synthetic data and validated the proposed model with four datasets including clinical applications. We guess the reviewer might misinterpret the comparison between different settings of our model (e.g.Bad-GAN (semi) + Good-GAN vs. Taichi-GAN in Table 1) as the comparison of our model and other baselines. Actually, our introduced Orthogonal loss and Good-GAN in the Bad-GAN framework all indicate significant improvements across datasets, as in Table 1. In addition, we have illustrated the qualitative results (Figure 4, Figure 6) and three further ablation studies (Table 2-3, Figure 5) to analyze the sensitivity of our model.

---

### Author Response · Authors · 2021-11-21
**Overall summary of responses**

Thank you for reviewing this paper. We are grateful for the feedback of all reviewers. We have detailed responses for each reviewer. Here is a brief summary showing that we have addressed the following major points:

1. Adding experiments of CIFAR10. As suggested by multiple reviewers, we have included CIFAR10, please see results in Table 1 of the revision. Our model consistently outperforms the benchmark methods and supports our conclusion.

2. About dataset complexity. We agree that MNIST and FashionMNIST are relatively simple datasets, as they are straightforward to illustrate the proposed ideas. To challenge the proposed ideas, we also included two more complex clinical datasets (HAM10000 and BraTS20) for evaluation; the relatively low AUCs in Table 1 from benchmark methods indicate their complexity. In addition, as suggested by reviewers, we also added the experiments on CIFAR10 in the revision.

3. About requiring the availability of labeled anomalies. We categorize our experiments on each of the data collections into two scenarios, i.e., “No anomalies available” and “Few anomalies available”. Our proposed models cover both scenarios (not only when labeled anomalies are available) and present consistent superior results (Table 1-3).

4. About the lack of “one anomaly class vs. many normal classes” settings.  First, In Table 2, we have included exactly the setting in the first column of Table 2 (“5” as the outlier,  the remaining numbers as inliers), and the results indicate our model is also superior along with other settings. In fact, we used Table 2 to illustrate that the improvement from our model is generalizable to various settings in terms of inlier/outlier distributions. Second, since outliers are usually heterogeneous and inliers are relatively compact (ChandolaVarun et al., 2009; Pang et al., 2021a), we believe the setting of  “one class as inlier and other classes as anomalies” makes more sense, which is also indicated in Deep SSVD (Ruff et al., 2018), FCDD (Liznerski et al., 2021), Deep SAD (Ruff et al., 2020b), etc.

5. Concerns of novelty. Our model is not simply to combine two types of GANs as some reviewers are concerned about, but to intuitively generate better pseudo anomalies to improve the anomaly detection performances. The value lies in data efficiency. When no labeled anomaly is available, we push the pseudo anomalies to lie at the boundary of inlier distribution with a new mean-shifted orthogonal loss to avoid pseudo anomalies converging to limited patterns. When few labeled anomalies are available, we push the pseudo anomalies to be closer to true anomalies by integrating Good-GAN into a Bad-GAN framework. In both scenarios, we illustrate superior performances than other benchmark approaches. We consider the novelty lies in bringing such value of data efficiency with our proposed method. Discussion can be found in Sec. 5.

---

### Decision · Program_Chairs · 2022-01-20

**Decision:**

Reject

**Comment:**

This paper proposes a method that combines Bad-GAN and Good-GAN, in which Good-GAN learns to generate the anomalies while Bad-GAN reguralizes the anomaly pseudo anomalies at the boundary of inlier distribution. In addition, a new orthogonal loss is proposed to  regularize the generation of anomaly samples to be distributed evenly at the periphery of the training data. The proposed method is new and shows some improvement over existing methods.

However, there are some detailed technical concerns raised by reviewers. Some of the concerns still remain unresolved after the discussion. 1) The proposed method lacks a principled way to select hyperparameters. 2) The experimental setting is a bit simple to verify the effectiveness of the proposed method in challenging real world applications. Especially, there is no theoretical guarantee of the proposed method, empirical evaluation is the only way to show the effectiveness of the proposed method. 3) The overall performance improvement is not very significant compared to existing methods. For example, the performance is very close to a method F-AnoGAN published in 2019. Addressing the concerns needs a significant amount of work. Thus, I do not recommend acceptance of this paper.